# High viral suppression rates among PLHIV on dolutegravir who had an initial episode of viral non-suppression in Uganda September 2020–July 2021

Grace A. Namayanja[1‡]*, Juliana de Fatima Da Silva[2], Bill Elur[1], Pamela M. Nasirumbi[1], Elliot Raizes[2], Julius Ssempiira[1‡], Esther Nazziwa[1], Miriam Nabukenya[3‡], Isaac Sewanyana[3‡], Jennifer Balaba[4‡], Jonathan Ntale[1‡], Jackie Calnan[5‡], Estella Birabwa[6‡], Juliet Akao[7‡], Christina Mwangi[1‡], Mary Naluguza[1‡], Arthur Ahimbisibwe[8‡], Cordelia Katureebe[8‡], Susan Nabadda[3‡], Lisa Nelson[1‡], Emilio Dirlikov[1]

1 Division of Global HIV and TB, US Centers for Disease Control and Prevention, Kampala, Uganda, 2 Division of Global HIV and TB, US Centers for Disease Control and Prevention, Atlanta, Georgia, United States of America, 3 Central Public Health Laboratories, Ministry of Health, Kampala, Uganda, 4 Monitoring and Evaluation Technical Support, Makerere University School of Public Health, Kampala, Uganda, 5 Office of Health and HIV, United States Agency for International Development, Kampala, Uganda, 6 Walter Reed Army Institute of Research, US Mission, Kampala, Uganda, 7 United States Department of Defense, US Mission, Kampala, Uganda, 8 AIDS Control Program, Ministry of Health, Kampala, Uganda

☯ These authors contributed equally to this work.
‡ JS, MN, IS, JB, JN, JC, EB, JA, CM, MN, AA, CK, SN, and LN also contributed equally and GAN first author and contributed equally to this work.
* uwo9@cdc.gov

**Data Availability Statement:** All relevant data are within the manuscript and its Supporting Information files.

## Abstract

### Background

In 2019, WHO recommended dolutegravir (DTG) as a backbone for first- and second-line antiretroviral therapy (ART) regimens for people living with HIV (PLHIV). According to the 2018 Uganda's HIV treatment guidelines, patients with viral non-suppression ($\geq$1,000 copies/mL) should receive intensive adherence counseling (IAC) with repeat viral load (VL) within 6 months. This analysis focused on the prevalence and factors associated with viral suppression following IAC among PLHIV on DTG-based regimens (DBRs) with an initial episode of viral non-suppression (VNS) in Uganda.

### Methods

We conducted a retrospective analysis for PLHIV on DBRs with an initial episode of VNS ($\geq$1,000 copies/mL) in Uganda during October 2019–September 2020 who had a follow up VL test result during September 2020–July 2021. Data were abstracted from the Central Public Health Laboratory (CPHL) database, including patient demographics and VL results. Viral non-suppression (VNS) was defined as a VL test result of $\geq$1,000 copies/mL. We characterized PLHIV on DBRs and used logistic regression models to determine factors associated with VL suppression after an initial episode of VNS.

**Funding:** Authors received no specific funding for this work.

**Competing interests:** The authors have declared no competing interest exist.

## Results

A total of 564 PLHIV on DBRs with an initial episode of VNS were followed up and 43 were excluded due to missing data. Of the 521, 220 (42.2%) were children (<15 years) and 231 (44.3%) were female. Median age was 28 years (interquartile range [IQR]: 12–43 years), and median duration on DBRs was 12 months (IQR: 6–15 months). Overall, 80.8% (421/521) PLHIV had a suppressed viral load at first follow up testing (children = 74.5% [164/220]; adults = 85.4% [257/301]). Children with initial VL results ≥5,000 copies/mL were less likely to achieve viral suppression at follow up testing compared to those with <5,000 copies/mL (AOR: 0.38; 95% CI: 0.20–0.71; p = 0.002).

## Conclusions

In a programmatic setting, most adults and children suppressed following an initial episode of VNS on DBRs. High rates of suppression after VNS suggest adherence challenges, rather than drug resistance. Continuation of DBRs should be considered before regimen switch.

## Introduction

Antiretroviral therapy (ART) reduces morbidity and mortality among people living with HIV (PLHIV) [1] and prevents transmission once viral load (VL) is below 200 viral copies/mL [2, 3]. Since 2019, the World Health Organization (WHO) has recommended VL testing as the preferred monitoring approach to assess treatment effectiveness [4]. At the patient level, VL testing is conducted at ≥6 and 12 months of starting a new ART regimen, and annually thereafter for those with viral suppression, generally defined as VL results <1,000 copies/mL [4]. Viral non-suppression (VNS) is indicated by results of ≥1,000 copies/mL, and virologic failure occurs upon two consecutive VNS test results taken within ≥3 months apart, with adherence support following the first VL test indicating VNS [1]. At the population level, achieving 95% VL suppression among PLHIV on ART is a key component of the UNAIDS strategy to end HIV by 2030 [5].

Efforts to achieve VL suppression target the two primary reasons for virologic failure: poor treatment adherence and lowered treatment effectiveness due to HIV drug resistance. WHO recommends intensified adherence counseling (IAC) as well as repeat VL testing for patients with VNS at the first test before treatment switch [4]. For patients with HIV drug resistance, treatment optimization is warranted to ensure ART effectiveness. Since 2019, WHO has recommended dolutegravir-based regimens (DBRs) as preferred first- and second-line ART (e.g., dolutegravir/lamivudine/tenofovir [TLD]), given dolutegravir's (DTG) potency, higher tolerability, low drug-drug interactions, and lower costs [6].

In 2018, the Uganda Ministry of Health (MOH) introduced DBRs as the preferred first-line ART regimens [7]. By September 30, 2022, over 1.31 million PLHIV were receiving ART, of whom 1.26 million (96%) were on DBRs; the overall VL suppression rate was 94% [8]. Once VL results are returned at the health facilities, PLHIV with VNS are called back to be informed of their result outcome and start IAC. Sessions for IACs are held at 1-month intervals, with repeat VL test done 1 month following the third IAC session. If viral suppression is not achieved, ART regimen switch to an optimal regimen is considered with an aim to achieve subsequent VL suppression [9]. Approaches of IAC in Uganda focus on 1) assessing the adherence barrier and client adherence practices; 2) advising the client on benefits of ART and

consequences of non-adherence; 3) assisting the client to identify the possible root cause of their non-adherence and identify strategies to overcome the barrier; 4) agreeing and documenting an adherence plan; and 5) arranging for the next appointment. DBRs are expected to have a very high barrier to drug resistance; switching patients from DBRs to protease inhibitors results in higher cost and pill burden with possible lower adherence.

It is therefore critical to provide patient-centered interventions to avoid unnecessary regimen switches. As part of the transition to DBRs in Uganda, we analyzed prevalence and factors associated with viral suppression following IAC among PLHIV on DBRs with an initial episode of VNS.

## Materials and methods

We conducted a retrospective analysis using deidentified data from the laboratory information system (LIMS) or database at Central Public Health Laboratory for PLHIV receiving U.S. President's Emergency Plan for AIDS Relief (PEPFAR)-supported ART on DBRs for $\geq 6$ months in Uganda with an initial episode of VNS ($\geq 1,000$ copies/mL) during October 2019–September 2020 and had a repeat VL test result during October 2020–July 2021. The DBRs were: tenofovir/lamivudine/dolutegravir (TDF/3TC/DTG); abacavir/lamuvidine/dolutegravir (ABC/3TC/DTG); zidovudine/lamivudine/dolutegravir (AZT/3TC/DTG); and salvage or third-line ART regimens given in case of resistance to non-nucleoside reverse transcriptase inhibitors (NNRTIs) and/or protease inhibitors (PIs). Data were abstracted from the CPHL electronic database. CPHL routinely conducts VL testing for the majority of PLHIV receiving ART supported by PEPFAR and captures deidentified patient data submitted through the national VL requisition form along with associated VL results (e.g., sex, date of birth, reason for VL testing, sample collection date). Since DTG had recently been rolled out and VL requisition forms had not been updated to capture the date of DBR initiation or transition, DBR start date was sourced from patient records at health facilities by the monitoring and evaluation staff from PEPFAR implementing partners, using the patient ART number from the VL requisition form. The VL requisition form also collects self-reported ART adherence assessment; those who reported to have taken all their pills or missed just one dose in the previous 1 month were classified as having good adherence (>95%); those who reported missing 2–4 doses were classified as having fair adherence (85–95%); and those who reported missing $\geq 5$ doses were classified as having poor adherence (<85%). IAC completion is documented in the high viral load register at the health facilities and not on the VL requisition form although it is required per the national HIV treatment guidelines [10]. Viral suppression was defined as VL test result <1,000 copies/mL for the VL test conducted during October 2020–July 2021 with prior VL results $\geq 1,000$ copies/mL during October 2019–September 2020. We reviewed PLHIV on DBRs for $\geq 6$ months who had VNS during October 2019–September 2020 and had a repeat VL test during October 2020–July 2021.

### Sample size estimation

From the CPHL database, there were 15,990 PLHIV on DBRs for $\geq 6$ months who had VNS during October 2019–September 2020 and had a repeat VL test during October 2020–July 2021. In order to ensure we could collect DBR start date from facilities, we used a simple random sampling, applying the below formula.

$$\mathbf{n} = \left(\left(\mathbf{pqZ}(1 - \alpha/2)^{\wedge}\mathbf{2}\right)/\mathbf{d}^{\wedge}\mathbf{2}\right)/\mathrm{mf}; \; n = \frac{pqZ_{1-\alpha/2}^2}{d^2}/mf$$

With a desired precision of 0.05 and an assumed non-response (incomplete data) rate of

8% from the facilities, a sample size was calculated separately for the children (aged <15 years) and adults (aged ≥15 years). We used the sample size calculation method for cross sectional studies utilizing the single proportion [11] and an application of a finite population correction factor for small populations (i.e., n/N ≥0.05, where n was the calculated sample and N was the known sampling frame), where an mf of 8% was used to adjust for non-response (incomplete data) by the identification numbers in the VL requisition forms and the database, and n/N>0.05 was adjusted to n/(1+n/N) [11].

## Data analysis

We analyzed data by sex, age (adults vs. children), region, DBR, health center type (i.e., health centers, hospitals, specialized clinics [e.g., HIV Treatment Centers of Excellence]), adherence level, VL results at initial and at repeat or follow up testing. For univariate analysis, we used frequency tables and summary statistics for the socio-demographic, clinical characteristics overall and group-specific variables. For bivariate analysis we used chi-square and t-tests to assess association between the independent variables and the outcome variable (viral suppression status) and for presence of association, a p-value of ≤0.05 was considered significant. Logistic regression was used to identify factors independently associated with viral suppression after prior non-suppression; unadjusted odds ratios (OR), adjusted odds ratios (AOR), and 95% confidence intervals (CI) were used to summarize effect measures.

## Ethical considerations

Given the use of data collected through routine clinical encounters, this analysis was conducted using a broad protocol that was approved by the U.S. Centers for Disease Control and Prevention (CDC) Global Health Center, Atlanta, Georgia as a non-research project (CDC Uganda broad protocol for use of routinely collected data for program improvement; project Id: 0900f3eb81a9093b). This broad protocol also received institutional review board exemption from the Makerere University School of Public Health Research and Ethics Committee, which served as a "waiver of consent." The investigators did not have access to personal identifying information from the CPHL database during or after data collection period. All the collected data were anonymous.

## Results and discussion

From October 2019 to September 2020, a total of 1,052,366 VL test results for samples collected from PLHIV who were on ART for ≥6 months were tested at CPHL. Of these 88,949 (8.5%), had VNS and had a subsequent VL test result during October 2020–July 2021. Among those who had VNS, 15,990 (18.0%) were on DBRs (864 children; 15,126 adults), of whom 564 (3.5%) were randomly selected to retrieve DBR start date and adherence information from the health facilities (Fig 1). Among the 564 selected, documented DBR start date was available for 521 (92.4%); of these, 510 (97.9%) had documented adherence assessment at initial VL testing and 493 (4.6%) had an adherence assessment at repeat testing (Fig 1).

## Demographics of PLHIV on dolutegravir-based regimens enrolled in the study

From the 521 PLHIV included in the analysis, 231 (44.3%) were female; 220 (42.2%) were children; and median age was 28 years (interquartile range [IQR]: 12–43 years). By geographic region, 38.2% (199/521) were in the Central Region, 28.2% (147/521) in the Northern Region, 26.7% (139/521) in the Western Region, and 6.9% (36/521) in the Eastern Region. By DBR

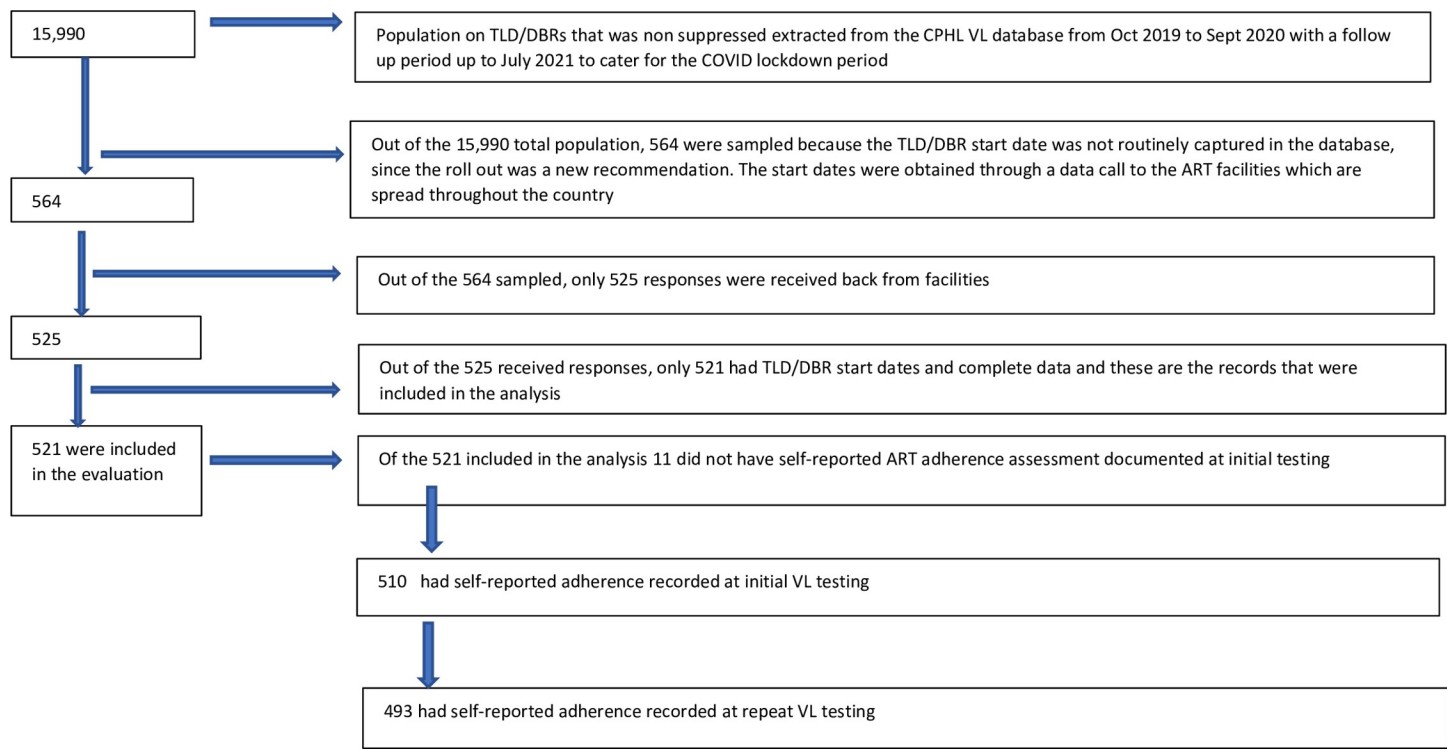

**Fig 1. Flow chart that shows how PLHIV were identified for inclusion in the study.**

regimen, 417 (80.0%) PLHIV received TDF/3TC/DTG, 97 (18.6%) were on ABC/3TC/DTG, 5 (1.0%) received AZT/3TC/DTG, and 2 (0.4%) salvage or third-line ART. The median duration on DBRs was 12 months (IQR: 6–15 months); it was 12 months (IQR: 8–18) for adult men, 12 months (IQR: 8–17 months) for adult women, and 9.9 months (IQR: 6–12 months) for children. Most PLHIV (73.7% [384/521]) were receiving ART from health centers, while 15.6% (81/521) and 10.7% (56/521) were receiving ART from hospitals and specialized clinics (e.g., HIV Treatment Centers of Excellence), respectively (Table 1).

## Median viral load for various characteristics of PLHIV with initial viral non-suppression

The median initial VNS result was 2,510 copies/mL (IQR: 1,420–12,070); 64.7% (337/521) had 1,000–4,999 copies/mL, 8.4% (44/521) had 5,000–9,999 copies/mL, and 26.9% (140/521) had ≥10,000 copies/mL. Median initial VNS result was 2,074copies/mL (IQR: 1,317–6,522) for adult men, 2,140 copies/mL (IQR: 1,430–8,580) for adult women, and 3,325 copies/mL (IQR: 1,482–21,313) for children. Median initial VNS result was 2,074 copies/mL (IQR: 1,359–6,954) for those on TDF/3TC/DTG, 2,558 copies/mL (IQR: 2,259–19,200) for ABC/3TC/DTG, and 16,726 copies/mL (IQR: 6,365–129,926) for the two clients on salvage or third-line ART. Median initial VNS result was 1,840 copies/mL (IQR: 1,312–3,781) for PLHIV receiving ART at health centers, 3,106 copies/mL (IQR: 1,780–59,245) for PLHIV receiving ART at hospitals, and 17,500 copies/mL (IQR: 3,610–182,000) for PLHIV receiving ART at specialized clinics (Table 1).

**Table 1. Demographics of PLHIV on dolutegravir-based regimens who had an initial high VL test result from October 2019–September 2020 and a follow up VL outcome from October 2020–July 2021 in Uganda.**

| Characteristic | Overall | | | | Suppressed | | |
|---|---|---|---|---|---|---|---|
| | Total, n(%) (N = 521) | Children, n(%) (N = 220) | Adults, n(%) (N = 301) | p-value | Children | Adults | p-value |
| **Sex** | | | | | | | |
| Male | 290 (55.7) | 120 (54.6) | 170 (56.5) | 0.661 | 87 (53.0) | 144 (56.0) | 0.549 |
| Female | 231 (44.3) | 100 (45.4) | 131 (43.5) | | 77 (47.0) | 113 (44.0) | |
| **Age Group** | | | | | | | |
| 0–9 | 51 (9.8) | 51 (23.2) | | | 39 (23.8) | | |
| 10–14 | 169 (32.4) | 169 (76.8) | | | 125 (76.2) | | |
| 15–24 | 28 (5.4) | | 28 (9.3) | | | 23 (9.0) | |
| ≥25 | 273 (52.4) | | 273 (90.7) | | | 234 (91.0) | |
| Median (IQR) | 28 (12–43) | 12 (10–13) | 41 (33–47) | | 12 (10–13) | 41 (33–48) | |
| **Health facility** | | | | | | | |
| HC IVs HC IIIs / HC II | 384 (73.7) | 148 (67.3) | 236 (78.4) | 0.015 | 106 (64.6) | 202 (78.6) | 0.006 |
| Hospital | 81 (15.6) | 41 (18.6) | 40 (13.3) | | 33 (20.1) | 34 (13.2) | |
| Centers of Excellence/Specialized HIV Clinics | 56 (10.7) | 31 (14.1) | 25 (8.3) | | 25 (15.2) | 21 (8.2) | |
| **Health facility ownership** | | | | | | | |
| Government (Public) | 379 (72.7) | 158 (71.8) | 221 (73.4) | 0.685 | 114 (69.5) | 188 (73.1) | 0.419 |
| Private Not For Profit (PNFP) | 142 (27.3) | 62 (28.2) | 80 (26.6) | | 50 (30.5) | 69 (26.9) | |
| **Regions of Uganda** | | | | | | | |
| Central | 199 (38.2) | 86 (39.1) | 113 (37.5) | 0.018 | 70 (42.7) | 98 (38.1) | 0.211 |
| East | 36 (6.9) | 11 (5.0) | 25 (8.3) | | 9 (5.5) | 21(8.2) | |
| North | 147 (28.2) | 75 (34.1) | 72 (23.9) | | 47 (28.7) | 60 (23.4) | |
| West | 139 (26.7) | 48 (21.8) | 91 (30.2) | | 38 (23.2) | 78 (30.4) | |
| **Duration on DTG in months** | | | | | | | |
| ≤6 months | 155 (29.8) | 101 (46.1) | 54 (17.9) | <0.001 | 80 (48.8) | 44 (17.12) | <0.001 |
| 7–12 months | 209 (40.2) | 81 (37.0) | 128 (42.5) | | 56 (34.2) | 113 (43.97) | |
| 13–24 months | 135 (26.0) | 30 (13.7) | 105 (34.9) | | 23 (14.0) | 87 (33.85) | |
| >24 months | 21 (4.0) | 7 (3.2) | 14 (4.7) | | 5 (3.0) | 13 (5.06) | |
| Duration on DTG regimen months (Median, IQR) | 12 (6–15) | 9.9 (6–12) | 13.5 (8–18) | | 9.7 (6–12) | 13.7 (8–18) | |
| **Self-reported ART Adherence levels at initial (high) VL testing N = 510** | | | | | | | |
| Poor (<85) | 3 (0.6) | 1 (0.5) | 2 (0.7) | 0.934 | 1 (0.6) | 2 (0.8) | 0.733 |
| Fair (85–95) | 27 (5.3) | 12 (5.6) | 15 (5.1) | | 11 (6.8) | 12 (4.8) | |
| Good (>95) | 480 (94.1) | 203 (94.0) | 277 (94.2) | | 149 (92.6) | 236 (94.4) | |
| **Self-reported ART Adherence levels at repeat VL testing (outcome) N = 493** | | | | | | | |
| Poor (<85) | 7 (1.4) | 2(1.0) | 5 (1.7) | 0.807 | 2 (1.3) | 3 (1.2) | 0.670 |
| Fair (85–95) | 33 (6.7) | 13(6.3) | 20 (7.0) | | 8 (5.3) | 19 (7.8) | |
| Good (>95) | 453 (91.9) | 191(92.7) | 262 (91.3) | | 142 (93.4) | 222 (91.0) | |
| **Viral load level at time of initial VNS episode (copies/mL)** | | | | | | | |
| ≥1,000–4,999 | 338 (64.9) | 127 (57.7) | 211 (70.1) | 0.003 | 104 (63.4) | 185 (72.0) | 0.065 |
| ≥5,000 | 183 (35.1) | 93 (42.3) | 90 (29.9) | | 60 (36.6) | 72 (28.0) | |
| Median (IQR) | 2,510 (1,420–12,070) | 3,325 (1,482–21,313) | 2,117 (1,360–7,700) | | 2,655 (1,417–15,951) | 2,040 (1,345–6,522) | |
| **Viral load level at repeat testing (VL outcome) copies/mL** | | | | | | | |

*(Continued)*

**Table 1.** (Continued)

| Characteristic | Overall | | | p-value | Suppressed | | p-value |
|---|---|---|---|---|---|---|---|
| | Total, n(%) (N = 521) | Children, n(%) (N = 220) | Adults, n(%) (N = 301) | | Children | Adults | |
| <1,000 (suppressed) | 421 (80.8) | 164 (74.6) | 257 (85.4) | 0.002 | 164 (100) | 257 (100) | |
| ≥1,000 (non-suppressed) | 100 (19.2) | 56 (25.5) | 44 (14.6) | | 0 (0.0) | 0 (0.0) | |
| Median (IQR) | 50 (0–839) | 110 (0–1,053) | 0 (0–839) | | 0 (0–815) | 0 (0–144) | |
| **Disaggregated Suppressed viral load level at repeat testing (copies/mL)** | | | | | | | |
| 0–49 | 258 (49.5) | 90 (40.9) | 168 (55.8) | | 90 (54.9) | 168 (65.4) | |
| 50–199 | 51 (9.8) | 26 (11.8) | 25 (8.3) | | 26 (15.9) | 25 (9.7) | |
| 200–499 | 8 (1.5) | 5 (2.3) | 3 (1.0) | | 5 (3.1) | 3 (1.2) | |
| 500–999 | 104 (20.0) | 43 (19.6) | 61 (20.3) | | 43 (26.2) | 61 (23.7) | |
| **DTG combinations** | | | | | | | |
| TDF/3TC/DTG | 417 (80.0) | 125 (56.8) | 292 (97.0) | | 95 (57.9) | 250 (97.3) | |
| ABC/3TC/DTG | 97 (18.6) | 92 (41.8) | 5 (1.7) | | 67 (40.9) | 5 (2.0) | |
| AZT/3TC/DTG | 5 (1.0) | 1 (0.5) | 4 (1.3) | | 1 (0.6) | 2 (0.8) | |
| Salvage: DTG/DRV and DTG/ETV/DRV/r | 2 (0.4) | 2 (0.9) | 0 (0.0) | | 1 (0.6) | 0 (0.0) | |

## Characteristics of PLHIV with viral load suppression at repeat testing

At repeat VL testing, 80.8% (421/521) achieved viral suppression (male = 79.7% [231/290]; female = 82.3% [190/231]; adults = 85.4% [257/301]; children = 74.5% [164/220]). By region, 84.4% (168/199) of PLHIV in the Central Region suppressed, 83.5% (116/139) in the Western Region, 83.3% (30/36) in the Eastern Region, and 72.8% (107/147) in the Northern Region. Viral suppression rates by DBR were: 82.7% (345/417) for TDF/3TC/DTG; 74.2% (72/97) for ABC/3TC/DTG; 60% (3/5) for AZT/3TC/DTG; and 50% (1/2) for salvage or third-line ART. Health facilities of different levels had similar suppression rates (health centers = 80.2% [308/384]; hospitals = 82.7% [67/81]; specialized clinics = 82.1% [46/56]). Among those who suppressed, the median duration on DBRs was longer for adults compared to children (adults = 13.7 months [IQR: 8–18 months] vs. children = 9.7 months [IQR: 6–12]; p <001) (Table 1).

## Self-reported adherence by PLHIV at initial and repeat VL testing

Overall, at initial VL testing, 97.9% (510/521) PLHIV on DBRs had a documented adherence assessment and of these, 94.1% (480/510) had good self-reported adherence despite VNS. Ninety-four percent (203/216) of children and 94.2% (277/294) of adults reported good adherence. Among 493 PLHIV who had a documented adherence assessment at repeat testing, 91.9% (453/493) had good self-reported adherence, 6.7% (33/493) had fair self-reported adherence, 1.4% (7/493) had poor self-reported adherence; 93.2% (191/205) children and 91.3% (262/287) adults reported good adherence at repeat testing. For those who were VNS at repeat testing, 25.7% (49/191) children and 15.3% (40/262) adults reported good adherence; 38.5% (5/13) children and 5.0% (1/20) adults reported fair adherence and 40.0% (2/5) adults reported poor adherence (Table 1).

## Factors associated with viral suppression

Children with VL ≥ 5000 copies/mL at initial VL test were less likely to suppress than those with 1,000–4,999 copies/mL (AOR: 0.38 95% CI [0.20–0.71]; p = 0.002). Compared to children in the Central Region, those in the Northern Region were less likely to suppress (OR: 0.38 95%

CI [0.19–0.79]), while those in the Eastern Region (OR: 1.03 95% CI [0.20–5.23] p = 0.973) and Western Region (OR: 0.87 95% CI [0.36–2.10] p = 0.755) had a similar likelihood to suppress (Table 2). There was no statistical difference in viral suppression between adults with <5,000 copies/mL or those with ≥5,000 copies/mL (AOR: 0.51 95% CI [0.26–1.01]). Other characteristics (i.e., sex, facility type, duration on DTG, and self-reported adherence) were not associated with viral suppression (Table 3).

## Discussion

In Uganda, we observed high rates of viral suppression among PLHIV on DBRs who had an initial episode of VNS. This finding suggests that the initial VNS episode likely resulted from poor adherence and that continuation of DBRs is an effective strategy for most patients for those with an initial VNS episode. Our findings are similar to a study by Buju R.T et al 2022 conducted in Bunia, Democratic Republic of Congo, where 72.8% of PLHIV on DBRs suppressed [12] and the ADVANCE trial (2021) that found high rates of suppression among PLHIV on DBRs, especially those on TDF/3TC/DTG [13]. Prior to the roll out of DBRs, Uganda's standard of care for PLHIV on NNRTIs, it was recommended to have immediate empirical switch to second-line regimens due to high rates of suspected drug resistance among those

**Table 2. Children: Correlation of demographic characteristics with odds of viral load suppression for those on dolutegravir based regimens.**

| Characteristic | Children % (n/N) N = 220 | Unadjusted odds ratios (95% CI) | P-value | Adjusted odds ratios (95% CI) | P-value |
|---|---|---|---|---|---|
| **Sex** | | | | | |
| Male | 120 (54.6) | 0.79 (0.43–1.46) | 0.446 | | |
| Female | 100 (45.5) | 1 | | | |
| **Health facility** | | | | | |
| Health centers | 148 (67.3) | 1 | | | |
| Hospitals | 41 (18.6) | 1.63 (0.70–3.83) | 0.258 | | |
| COE/Specialist clinics | 31 (14.1) | 1.65 (0.63–4.31) | 0.306 | | |
| **Type of health facility ownership** | | | | | |
| Government (Public) | 158 (71.8) | 0.62 (0.30–1.28) | 0.196 | | |
| Private Not For Profit (PNFP) | 62 (28.2) | 1 | | | |
| **Region** | | | | | |
| Central | 86 (39.1) | 1 | | | |
| East | 11 (5.0) | 1.03(0.20–5.23) | 0.973 | | |
| North | 75 (34.1) | 0.38 (0.19–0.79) | 0.009 | 0.08 (0.61–1.04) | 0.099 |
| West | 48 (21.8) | 0.87 (0.36–2.10) | 0.755 | | |
| **Duration on DTGs** | | | | | |
| ≤1 year | 182 (83.1) | 1 | | | |
| >1 year | 37 (16.9) | 1.05 (0.46–2.39) | 0.903 | | |
| **Self-reported good ART adherence levels at time of initial VNS** | | | | | |
| Yes (>95) | 203 (94.0) | 0.23 (0.03–1.81) | 0.163 | | |
| No (<95) | 13 (6.0) | 1 | | | |
| **Self-reported good ART adherence levels at time of repeat VL testing** | | | | | |
| Yes (>95) | 191 (92.7) | 1.45 (0.47–4.45) | 0.517 | | |
| No (<95) | 15 (7.3) | 1 | | | |
| **High viral load at time of initial VNS** | | | | | |
| 1,000–4999 | 127 (57.7) | 1 | | | |
| ≥5,000 | 93 (42.3) | 0.04 (0.22–0.75) | 0.004 | 0.38 (0.20–0.71) | 0.002 |

**Table 3. Adults: Correlation of demographic characteristics with odds of viral load suppression for those on dolutegravir-based regimens.**

| Characteristic | Adults % (n/N) N = 301 | Unadjusted odds ratios (95% CI) | P-value | Adjusted odds ratios (95% CI) | P-value |
|---|---|---|---|---|---|
| **Sex** | | | | | |
| Male | 170 (56.5) | 0.88 (0.46–1.69) | 0.705 | | |
| Female | 131 (43.5) | 1 | | | |
| **Health facility** | | | | | |
| Health centers | 236 (78.4) | 1 | | | |
| Hospitals | 40 (13.3) | 0.95 (0.37–2.44) | 0.921 | | |
| COE/Specialist clinics | 25 (8.3) | 0.88 (0.29–2.73) | 0.830 | | |
| **Type of health facility ownership** | | | | | |
| Government (Public) | 221 (73.4) | 0.91 (0.44–1.90) | 0.798 | | |
| Private Not For Profit (PNFP) | 80 (26.6) | 1 | | | |
| **Region** | | | | | |
| Central | 113 (37.5) | 1 | | | |
| East | 25 (8.3) | 0.80 (0.24–2.67) | 0.721 | | |
| North | 72 (23.9) | 0.77 (0.34–1.75) | 0.525 | | |
| West | 91 (30.2) | 0.92 (0.41–2.04) | 0.835 | | |
| **Duration on DTGs** | | | | | |
| ≤1 year | 182 (60.5) | 1 | | | |
| >1 year | 119 (39.5) | 0.84 (0.44–1.60) | 0.593 | | |
| **Good ART adherence levels at initial VL testing** | | | | | |
| Yes (>95) | 277 (94.2) | 1.23 (0.34–4.48) | 0.750 | | |
| No (<95) | 17 (5.8) | 1 | | | |
| **Good ART adherence levels at repeat VL testing** | | | | | |
| Yes (>95) | 262 (91.3) | 0.76 (0.22–2.65) | 0.663 | | |
| No (<95) | 25 (8.7) | 1 | | | |
| **High viral load at initial testing** | | | | | |
| 1000–4999 | 287 (95.4) | 1 | | | |
| ≥5,000 | 14 (4.7) | 0.56 (0.29–1.09) | 0.087 | 0.51 (0.26–1.01) | 0.052 |

with virologic failure. This recommendation is contrasted by high suppression among PLHIV on DBRs suppression rates, as seen in a trial context like the ADVANCE as well as in the real-world per our analysis of program data, which supports WHO recommendation of enhanced adherence counseling following VNS instead of prompt empirical regimen switch to more costly regimens that are often not available in fixed-dose combinations, potentially further negatively impacting adherence.

We recognize that even though the viral suppression rate was high, almost one in five patients with an initial episode of VNS failed to suppress. We infer that these individuals might require additional patient-centered interventions to address the underlying factors of lower adherence or further investigations to uncover the reasons for their non-VNS, including burgeoning drug resistance. There are few data in sub-Saharan Africa on prevalence of dolutegravir drug resistance at the time of virologic failure. More data are needed to guide programmatic management among those who fail to suppress following VNS.

## Non-suppression at initial VL testing for PLHIV on dolutegravir-based regimens

In this study, we observed high rates of self-reported adherence at initial consultation, with lower rates of self-reported adherence at follow-up. In programmatic settings with many

patients, it is difficult to accurately assess patient adherence. In Uganda, we rely on patient self-reported adherence, as communicated to the clinician and adherence counsellors, with additional support given for patients with VNS. During the initial clinical visits, patients might self-report good adherence, despite difficulties in following clinician instructions, for several reasons, including perceived negative repercussions and potential for a prolonged clinical encounter. Thus, viral load testing is an important tool for monitoring treatment effectiveness, through which additional patient-centered services can be initiated toward supporting improved treatment adherence. For example, patients with VNS interface with the adherence counselor through a one-on-one session (sometimes on a special day focus on non-suppressed patients), toward gaining insights into adherence barriers. Repeat IAC sessions help to further identify barriers and support patients with treatment adherence.

This analysis was conducted shortly after Uganda had rolled out DBRs in late 2018. Thus, patients had a median duration on DBRs of 1 year; suppression rates might differ among patients on DBRs for longer periods of time. Majority of the patients were male, reflecting alignment with initial guidance from WHO urging caution with DBR roll out, given potential signal of neuro-tube defect signal among infants born to women living with HIV taking DBRs [14]. PLHIV on AZT/3TC/DTG (a non-fixed-dose combination regimen) had the highest VL at initial testing, possibly due to the higher pill burden and the twice-daily dosing of AZT/3TC, which can present challenges for adherence [15]. Several health facilities in Uganda attach expert clients or peers to support those who disclose ART adherence challenges. This initiative has been helpful in other countries, as seen in a meta-analysis done by Berg [16]. Higher VNS at initial testing was observed among those receiving treatment from specialized clinics. This may be because patients enrolled there are usually referred from lower health facilities and are likely to have been on ART for a long time, developed complications requiring specialized management. In addition, these facilities are likely to be further away from the PLHIV residences and communities, and incurred transport costs to the facilities has been identified as a major factor contributing to ART non-adherence [17]. With support from PEPFAR, Uganda has scaled up differentiated service delivery models and taken services closer to communities at locations chosen by PLHIV with the goal to improve patient-centered care, client satisfaction, adherence, and treatment continuity.

### Repeat viral load testing following intensive adherence counseling

A study conducted in Gomba district in Central Uganda, showed high IAC completion of 81% for those with VNS [18]. We noted high VL suppression rates among those on TDF/3TC/DTG and ABC/3TC/DTG at repeat testing and this could possibly be due to once-a-day dosing compared to AZT/3TC/DTG and third line regimens that are usually taken twice daily and have higher pill burdens. This is further shown by the higher median VL for those on AZT/3TC/DTG compared to TDF/3TC/DTG. This is similar to what was found in a metanalysis where twice-daily regimens had unfavorable outcomes compared to once-daily regimens [15]. WHO recommends using TDF/3TC/DTG in both first- and second-line regimens, which simplifies and improves ART adherence [6].

### Children

In Uganda, DTG 10mg was rolled out in late 2021, after this analysis was concluded. During the analysis period, only children and adolescents weighing ≥20kg were eligible to take the available formulation of DTG 50mg, and therefore children had a shorter duration on DBRs compared to adults. Children had a higher VL at initial testing compared to adults possibly due to children relying on caregivers for whom high pill-burden regimens may be harder to

consistently administer to the children and they are taken twice a day. This finding is similar to a study done in Swaziland [19], where children and adolescents had lower suppression rates following IAC. Introduction of fixed-dose combination ABC/3TC/DTG for children may improve adherence and suppression rates. In this study, we observed that many children in the Northern Region were less likely to suppress compared to those in the Central Region, which could be attributable to the recovery from armed conflict in the region [20] and the resulting phased roll out of DBRs used as part of the ART optimization process in that region [21]. Overall, these findings highlight that additional adherence support for children is needed. One such program, the Youth Adherence Peer Support (YAPS) program, utilizes peer-to-peer counseling and linkage to orphan and vulnerable children services that offer food and transport to children in vulnerable households to boost ART adherence [22] and VL suppression.

## Limitations

There are several limitations to this study. First, we used patient-level data collected through routine clinical encounters and as documented in the VL requisition form, which might be prone to data quality issues that influenced results. For example, we acknowledge that we noted that a small proportion of PLHIV did not have adherence assessments at initial and repeat testing with this routinely collected data. However, compared to data collected through study investigations, these data more accurately reflect real-world, programmatic settings. Moreover, use of these data allowed us to identify children and adults throughout the country with an initial episode of VNS. Second, we assumed that all patients had three IAC sessions between the two viral load tests, as per the national guidelines, although documentation of IAC sessions can be incomplete across data sources. Furthermore, given the limitations of the CPHL LIMS database, we were not able to fully assess if all criteria per the national guidelines were met for a repeat viral load test (i.e., three IAC sessions with >95% self-reported adherence) or conduct an analysis of the time between the two viral load tests per individual. Despite this limitation, we can be confident in our assumption, given continual training on the national guidelines, the need for three IAC to order a repeat viral load, and documentation requirements for further drug resistance testing if the patient experiences virologic failure. Third, adherence was self-reported by patients. More intensive investigations, such as direct-observation of treatment, would be needed to counter this potential bias, but was not feasible under programmatic conditions. Further, if we take self-reported adherence as accurate, then our results may have been biased to include patients with better adherence, given the criteria for a repeat viral load (i.e., three IAC sessions with >95% self-reported adherence). This would seem to strongly bias results towards finding viral suppression. Finally, analyzes were limited to variables available within the database, preventing sub-analyses, such as suppressions rates among those on first line DBRs compared to treatment experienced patients.

## Conclusions

In a programmatic setting, most adults and children suppressed following an initial episode of VNS. High rates of suppression suggest initial adherence challenges, rather than drug resistance. Continuation of DBRs may be helpful before switch to regimens with a higher pill burden which the PLHIV may not adhere to as well.

### Recommendations

These findings support the international recommendation of delaying regimen switch among PLHIV on DTG with an initial episode of VNS since the main reason for non-suppression for majority of PLHIV is non-adherence and not drug resistance. Therefore, immediate switches

to PIs with more pill burden will worsen their adherence and not improve their suppression. In addition, patients with high self-reported adherence are likely to benefit from routinely offered adherence services like what is given to those who have disclosed poor or fair adherence. More studies may be helpful to ascertain why children in northern Uganda were less likely to virally suppress. Routine drug resistance monitoring may be necessary for those who don't achieve viral suppression but have good adherence after IAC efforts have been exhausted. Programs may consider strengthening psychosocial support and adherence interventions, not regimen switch, as the preferred management strategy for patients failing DBRs in line with the WHO recommendation.

## Supporting information

**S1 Dataset. High viral suppression dataset_June 2024.**
(XLSX)

**S1 File. Inclusivity-in-global-research-questionnaire.**
(DOCX)

## Acknowledgments

The CPHL team that helped with data extraction, Fatuma Nalubega from METS, Norah Namuwenge from SITES, Implementing Partners and M&E teams from PEPFAR supported Implementing Partners who helped to provide the DTG start dates.

 **Disclaimer**:

 • "*The findings and conclusions in this [report/presentation] are those of the author(s) and do not necessarily represent the official position of the funding agencies.*

 • "*This analysis has been supported by the President's Emergency Plan for AIDS Relief (PEPFAR) through the Centers for Disease Control and Prevention (CDC) under the terms of . . .the Uganda Protocol for Use of Routinely Collected Data for Program Improvement.* Project Id: 0900f3eb81a9093b

## Author Contributions

**Conceptualization:** Grace A. Namayanja, Emilio Dirlikov.

**Data curation:** Miriam Nabukenya, Isaac Sewanyana, Jennifer Balaba.

**Formal analysis:** Bill Elur, Pamela M. Nasirumbi.

**Methodology:** Bill Elur, Pamela M. Nasirumbi.

**Supervision:** Esther Nazziwa.

**Validation:** Bill Elur, Julius Ssempiira, Miriam Nabukenya, Jennifer Balaba.

**Visualization:** Bill Elur, Julius Ssempiira.

**Writing – original draft:** Grace A. Namayanja, Isaac Sewanyana, Emilio Dirlikov.

**Writing – review & editing:** Grace A. Namayanja, Juliana de Fatima Da Silva, Elliot Raizes, Julius Ssempiira, Jonathan Ntale, Jackie Calnan, Estella Birabwa, Juliet Akao, Christina Mwangi, Mary Naluguza, Arthur Ahimbisibwe, Cordelia Katureebe, Susan Nabadda, Lisa Nelson, Emilio Dirlikov.

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
