## [Decision Letter · Decision Letter 0]

20 Dec 2023

PONE-D-23-22283High Viral Suppression Rates among PLHIV on Dolutegravir who had an Initial Episode of Viral Non-Suppression in Uganda September 2020–July 2021.PLOS ONE

Dear Dr. Namayanja,

Thank you for submitting your manuscript to PLOS ONE. After careful consideration, we feel that it has merit but does not fully meet PLOS ONE’s publication criteria as it currently stands. Therefore, we invite you to submit a revised version of the manuscript that addresses the points raised during the review process.

Kindly review the comments from the reviewers and revise the manuscript.

We look forward to receiving your revised manuscript.

Kind regards,

Chika Kingsley Onwuamah, Ph.D.

Academic Editor

PLOS ONE

Journal Requirements:

4. Please note that your Data Availability Statement is currently missing [the repository name and/or the DOI/accession number of each dataset OR a direct link to access each database]. If your manuscript is accepted for publication, you will be asked to provide these details on a very short timeline. We therefore suggest that you provide this information now, though we will not hold up the peer review process if you are unable.

Reviewers' comments:

Reviewer's Responses to Questions

**Comments to the Author**

1. Is the manuscript technically sound, and do the data support the conclusions?

Reviewer #1: Partly

Reviewer #2: Yes

2. Has the statistical analysis been performed appropriately and rigorously? 

Reviewer #1: No

Reviewer #2: I Don't Know

3. Have the authors made all data underlying the findings in their manuscript fully available?

Reviewer #1: No

Reviewer #2: No

4. Is the manuscript presented in an intelligible fashion and written in standard English?

Reviewer #1: Yes

Reviewer #2: Yes

5. Review Comments to the Author

Reviewer #1: The contribution of the paper is unclear. DBRs were introduced because they are effective. More is needed in terms of the factors associated with viral suppression. The analysis focuses on geography, type of facility, duration DBRs. It is not clear what lessons can be taken away and which may be used to inform programmes.

Reviewer #2: Namayanja et al. studied individuals treated for HIV with dolutegravir-based regimens who experienced an episode of viral non-suppression, then received intensive adherence counseling for 3 months, then were retested for viral load. The authors found that 80% of individuals had a suppressed viral load at followup, suggesting that their prior non-suppression was due to lower medication adherence rather than viral resistance. Overall, this is a well described study that makes important use of a large amount of data in the Uganda Central Public Health Laboratory. I have several questions that would help readers understand and interpret this information.

Please include information about the the followup interval for each individual, i.e. the amount of time between the initial episode of viral non-suppression and the first followup visit. Was the duration of that interval associated with suppression (in either direction)?

Why do the authors think that the reported adherence decreased, but viral suppression increased, at the first followup?

Please provide more detail about the formula used for sample size calculation, for a broad audience.

Please check line 160 – is this text intended to be in the manuscript?

Line 282 indicates that repeat viral load testing took place if the patient reported >95% medication adherence. This would seem to strongly bias results towards finding viral suppression. If this is accurate, I would include this earlier in the manuscript as a condition of the specific population under investigation (ie, those with good medication adherence).

6. PLOS authors have the option to publish the peer review history of their article (what does this mean?). If published, this will include your full peer review and any attached files.

Reviewer #1: No

Reviewer #2: No

---

## [Decision Letter · Decision Letter 1]

12 Mar 2024

PONE-D-23-22283R1High Viral Suppression Rates among PLHIV on Dolutegravir who had an Initial Episode of Viral Non-Suppression in Uganda September 2020–July 2021.PLOS ONE

Dear Dr. Namayanja,

Thank you for submitting your manuscript to PLOS ONE. After careful consideration, we feel that it has merit but does not fully meet PLOS ONE’s publication criteria as it currently stands. Therefore, we invite you to submit a revised version of the manuscript that addresses the points raised during the review process.

Some comments from the reviewers have been addressed. Kindly address the outstanding comments from a reviewer, which has been reiterated. These are critical to the study methodology and findings.

We look forward to receiving your revised manuscript.

Kind regards,

Chika Kingsley Onwuamah, Ph.D.

Academic Editor

PLOS ONE

Reviewers' comments:

Reviewer's Responses to Questions

**Comments to the Author**

1. If the authors have adequately addressed your comments raised in a previous round of review and you feel that this manuscript is now acceptable for publication, you may indicate that here to bypass the “Comments to the Author” section, enter your conflict of interest statement in the “Confidential to Editor” section, and submit your "Accept" recommendation.

Reviewer #1: All comments have been addressed

Reviewer #2: (No Response)

2. Is the manuscript technically sound, and do the data support the conclusions?

Reviewer #1: (No Response)

Reviewer #2: Partly

3. Has the statistical analysis been performed appropriately and rigorously? 

Reviewer #1: (No Response)

Reviewer #2: I Don't Know

4. Have the authors made all data underlying the findings in their manuscript fully available?

Reviewer #1: (No Response)

Reviewer #2: No

5. Is the manuscript presented in an intelligible fashion and written in standard English?

Reviewer #1: (No Response)

Reviewer #2: Yes

6. Review Comments to the Author

Reviewer #1: (No Response)

Reviewer #2: While the authors have addressed some of my prior comments, others have not yet been addressed and the text was not updated to include important information. It would also be helpful if the authors can explain what changes have been made to the tables.

Please include information about the the followup interval for each individual on a per-individual basis, i.e. the amount of time between the initial episode of viral non-suppression and the first followup visit. Was the duration of that interval associated with suppression (in either direction)? The authors replied that the visits were meant to be one month apart, but did not indicate specific information about the interval or evaluate whether the interval was associated with suppression. For example, if someone didn’t come back until 6 weeks later, were they more likely to be non-suppressed? Please also address this question in the manuscript itself, not just the response here.

Why do the authors think that the reported adherence decreased, but viral suppression increased, at the first followup? The authors replied that the paper shows “when ART adherence improves, the PLHIV get virally suppressed” – while I agree with this, the analysis also shows that viral suppression can happen before adherence improves..

At repeat VL testing, 80.8% (421/521) achieved viral suppression

Overall, at initial VL testing, … 94.1% (480/510) had good self-reported adherence despite VNS

Among 493 PLHIV who had a documented adherence assessment at repeat testing, 91.9% (453/493) had good self-reported adherence

What would explain that? Please also address this question in the manuscript itself, not just the response here

The section on line 160 (now line 176) does not seem appropriate for the main text of a manuscript, based on the way it is phrased and the way it describes individual investigators and explains why they are not co-authors. I haven’t seen something like this in the main text of a manuscript before. I recommend removing it and putting this information in acknowledgements, funding, etc. The specific text is pasted below, including the word “Response:” which seems to me like a response to a review question, and not something that should be in the manuscript.

Response: “CDC Uganda with support from the associate director of science, strategic information branch chief, CDC Uganda staff and Ministry of health among others developed a broad protocol that provides for review and use of retrospective routinely collected de-identified data. This protocol received approval from the CDC Science and integrity branch and exemption from the Makerere School of Public Health to develop and write abstracts manuscripts from various data sources used in the country e.g., dashboards, databases, Registers. In our manuscript we utilized de-identified data from the Central Public Health laboratory (CPHL) database or Lab Information management system (LIMS). The CPHL conducts viral load testing on all samples national wide. This broad protocol had the senior leadership from the various institutions that CDC works with e.g., MOH – where Dr. Joshua Musinguzi is the Program Manager of the AIDS control program and CDC Uganda Strategic Information branch chief – Jenny Ward. Joshua and Jenny only served as the co-authors on the CDC broad protocol for programmatic data utilization. However, for this manuscript they were not specifically engaged in the work and therefore, we did not include them in as co-authors.

Line 282 indicates that repeat viral load testing took place if the patient reported >95% medication adherence. This would seem to strongly bias results towards finding viral suppression. Please include this earlier in the manuscript as a condition of the specific population under investigation (ie, those with good medication adherence), and include this as an overall limitation to the study.

7. PLOS authors have the option to publish the peer review history of their article (what does this mean?). If published, this will include your full peer review and any attached files.

Reviewer #1: **Yes: **Akudo Ikpeazu MD, MPH, DrPH

Reviewer #2: No

---

## [Author Response · Author response to Decision Letter 1]

27 Apr 2024

I thank the reviewers for their time to make the paper better. I have uploaded the Revised Manuscript with Track changes, Manuscript_April 27, Response to reviewer comments, and the Data set with DO STAT files. The spreadsheet has various tabs for you to review as needed. I have removed the figure from the body of the manuscript. Hope to hear from you.

---

## [Decision Letter · Decision Letter 2]

24 May 2024

High Viral Suppression Rates among PLHIV on Dolutegravir who had an Initial Episode of Viral Non-Suppression in Uganda September 2020–July 2021.

PONE-D-23-22283R2

Dear Dr. Namayanja,

We’re pleased to inform you that your manuscript has been judged scientifically suitable for publication and will be formally accepted for publication once it meets all outstanding technical requirements.

Kind regards,

Chika Kingsley Onwuamah, Ph.D.

Academic Editor

PLOS ONE

Additional Editor Comments (optional):

Reviewers' comments:

Reviewer's Responses to Questions

**Comments to the Author**

1. If the authors have adequately addressed your comments raised in a previous round of review and you feel that this manuscript is now acceptable for publication, you may indicate that here to bypass the “Comments to the Author” section, enter your conflict of interest statement in the “Confidential to Editor” section, and submit your "Accept" recommendation.

Reviewer #2: All comments have been addressed

2. Is the manuscript technically sound, and do the data support the conclusions?

Reviewer #2: (No Response)

3. Has the statistical analysis been performed appropriately and rigorously? 

Reviewer #2: (No Response)

4. Have the authors made all data underlying the findings in their manuscript fully available?

Reviewer #2: (No Response)

5. Is the manuscript presented in an intelligible fashion and written in standard English?

Reviewer #2: (No Response)

6. Review Comments to the Author

Reviewer #2: (No Response)

7. PLOS authors have the option to publish the peer review history of their article (what does this mean?). If published, this will include your full peer review and any attached files.

Reviewer #2: No

---

## [Editor Report · Acceptance letter]

30 May 2024

PONE-D-23-22283R2 

PLOS ONE

Dear Dr. Namayanja, 

I'm pleased to inform you that your manuscript has been deemed suitable for publication in PLOS ONE. Congratulations! Your manuscript is now being handed over to our production team.

Kind regards, 

on behalf of

Dr. Chika Kingsley Onwuamah 

Academic Editor

PLOS ONE